# Lining Fatigue Test and Influence Zoning of Tridimensional Cross-Tunnel under High-Speed Train Loads

**Weichao Yang** [1]**, E Deng** [1,]*****, Chenghua Shi** [1]**, Ning Liu** [2]**, Ruizhen Fei** [1] **and Huan Yue** [1]

[1] School of Civil Engineering, Central South University, Changsha 410075, China; weic_yang@csu.edu.cn (W.Y.); csusch@163.com (C.S.); feiruizhen@csu.edu.cn (R.F.); 02160388@cumt.edu.cn (H.Y.)

[2] College of Civil Engineering, Guizhou University, Guiyang 550025, China; nliu1@gzu.edu.cn

***** Correspondence: denge12@csu.edu.cn; Tel.:+86-133-9749-8548



**Featured Application: The results of this paper can provide a further theoretical basis for the evaluation of lining fatigue damage in the tridimensional cross tunnel of high-speed railway.**

**Abstract:** Tridimensional cross tunnels usually manifest the vulnerable components of a high-speed railway caused by the sophistication of the structural pattern and the continuous shock from the train. The frequent defect of tunnel lining at the intersection would affect the safe operation of the two rails. As a result, attention has been paid to fatigue damage caused by the long-term dynamic load from a running train, in order to ensure the safety and serviceability of the cross tunnel lining. However, an influence zoning method with respect to tunnel crossing for the direct estimation of whether the lining structure is damaged due to the train load, and to what extent, is unavailable. In this paper, a systematic study that consists of numerical simulation and fatigue damage experiment is conducted to develop an approximate method to enable practicing engineers to evaluate reasonable design parameters. The initial static stress, which corresponds to the static tensile stress of secondary lining under the stratum load, and the maximum dynamic stress, which refers to the maximum dynamic tensile stress under the train load, are estimated according to the numerical simulation. A simplified damage evolution model and its parameters are identified on the basis of a systematic fatigue damage experiment. Finally, the influence zoning method is conducted on the basis of two criteria, namely (1) that initial stress level should not exceed 0.6, and (2) that load cycles should not exceed $N = 2 \times 10^6$ times. Thus, the practicing parameters during the cross tunnel design, such as surrounding rock mass, cross angle, rock pillar thickness between two tunnels, and train speed can be utilized conveniently by using the proposed calculation charts, according to the identification of initial stress level and the magnitude of dynamic stresses caused by the train load.

**Keywords:** tridimensional cross tunnel; lining; train loads; fatigue test; vibration response; influence zoning

## 1. Introduction

Given that the high-speed railway (HSR) in China has reached 30,000 km, the number of HSR tunnels under- or over-passing road tunnels [1], subway tunnels [2–5], another trunk railway tunnel [6,7], and even another HSR tunnel has been increasing. For example, the Huofengshan Tunnel of the Chongqing comprehensive transportation hub passes over the Renhechang Tunnel of the Chongqing–Huaihua Railway, and the minimum distance of the intersection is only 5.47 m. The Anhui and Jiangxi dual-line railway and the Jiujiang–Quzhou Railway have a tunnel crossover

in Jingdezhen, and the minimum distance of the intersection is 4.5 m. The Guantouling Tunnel of the Wenzhou–Fuzhou Railway passes under the Guantouling Expressway Tunnel, and the railway tunnel arch is approximately 2.91 m from the base of the Guantouling Tunnel. In addition, several cross tunnels are found in the Beijing–Guangzhou and the Shanghai–Kunming HSR.

The cross tunnel lining shares relatively higher stress levels compared with ordinary tunnels. Many scholars have investigated the interaction mechanism and engineering characteristics in cross tunnel construction to mitigate the adverse effect in cross-tunnelling construction. Cross-tunnelling increases the initial stress of the tunnel lining if the tunnel distance is close [8]. During the construction of an existing tunnel under a vertical tunnel, the longitudinal soil pressure of the existing surrounding rock of the tunnel has an "arched" distribution [9]. The stress of the upper tunnel lining changes consequently [10], the compressive failure of the shotcrete lining of the crown can occur, and the tensile forces of the rock bolts around the crown can increase substantially [11,12] because of the maximum additional stress (approximately 0.7 MPa) induced by shield tunnelling below [13]. Other investigations have also presented that the stress level of the tunnel lining and the surrounding rock in an intersection is generally higher than that in other locations [14–16].

The long-term train loading would be another potential threat to the serviceability of cross tunnels. At present, approximately 65% of tunnels in China suffer such variable damage, such as cracking, stripping, void, seepage, and even mud pumping [17,18]. The dynamic load from a running train could cause damage cracks in the tunnel linings and plastic deformation of surrounding rock, because of the high initial stress of the cross tunnels [19–21]. Moreover, the cross tunnel would increase the static stress level of the tunnel lining because of its complex structure [4,5,22]. The dynamic response of the tunnel would affect the durability and service performance of the lining because of the high static stress of the cross tunnels; moreover, the influence of a long-term dynamic load induced by high-speed trains (HSTs) should be considered in the design work of similar engineering cases [23].

This defect of the HSR cross tunnel, with a 100 year designed service duration, repeatedly occurs at an intersection, and the safe operation of the two rails would suffer an adverse effect. Thus, defining parameters, such as surrounding rock mass ($\lambda$), cross angle ($\theta$), rock mass thickness between tunnels ($H$), and train speed ($v$), is urgent and necessary in the design stage, especially for practicing engineers to directly estimate whether the lining structure is damaged because of the train load, and to what extent. In this paper, a systematic study that consists of numerical simulation and a fatigue damage experiment is conducted to develop an approximate method to enable practicing engineers to evaluate the reasonable design parameters of cross tunnels.

The framework of this paper is organised as follows. Firstly, the methodology, including numerical simulation and experiment, is presented (Section 2). Secondly, the results, such as the initial stress distribution of the secondary lining under the stratum load, the increment of dynamic tensile stress under the train load based on the numerical simulation, the evolution characteristics from fatigue experiment, and the damage model and its parameters, are outlined (Section 3). Thirdly, the influence zoning methods and the calculation charts that involve cross tunnels are provided (Section 4). The last part contains the main conclusions and recommendations (Section 5).

## 2. Methodology

The minimum and maximum fatigue stresses are necessary to investigate the damage of secondary lining caused by dynamic loading. The minimum fatigue stress, which is considered the initial static stress of the lining from the stratum load, and the maximum fatigue stress, which is equal to the accumulation of static and dynamic stresses caused by the running train, are investigated via numerical simulation in this study. Subsequently, a systematic fatigue experiment is executed to generalise a simplified damage evolution model and to regress its parameters.

### 2.1. Numerical Simulation

#### 2.1.1. Numerical Configuration

The analyses of static stress distribution and the dynamic stress increment were carried out using the finite MIDAS/GTS 12 in this paper. A typical tunnel prototype was deduced from the standard tunnel for an HSR double-line in accordance with the code TB10621-2014 (Figure 1). The length of the soil grid was 50 m from each left, right, front, and back portion along the upper tunnel up to the surface for all investigated cases. These cases were based on the results of a sensitivity analysis that was conducted to investigate potential boundary effects on the computed response at the central area of the numerical models. The surface was buried deep, according to the buried-deep criteria-that is, 32 m. The main factors influencing the dynamic response of the lining of the three-dimensional cross tunnel under the load of HSTs are the surrounding rock level ($\lambda$), the direction the train passes ($\kappa$), train speed ($v$), cross-angle ($\theta$), and the rock thickness between two tunnels ($H$). According to these factors, 13 numerical models were established. Among them, the surrounding rock of the V level, the train passing through the upper tunnel at a speed of 350 km/h, the rock pillar at 1 m, and a cross angle of 90° were determined to be the basic working conditions. The calculation model grid of the typical working condition is shown in Figures 2 and 3.

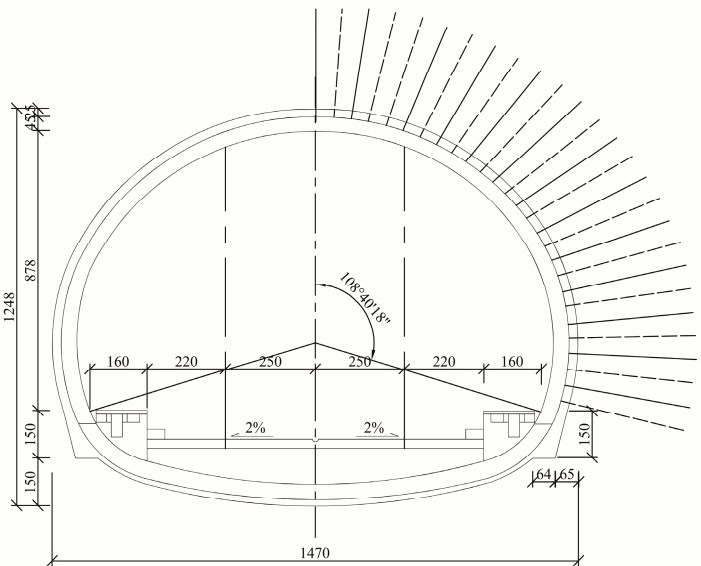

**Figure 1.** Layout of tunnel section (unit: cm).

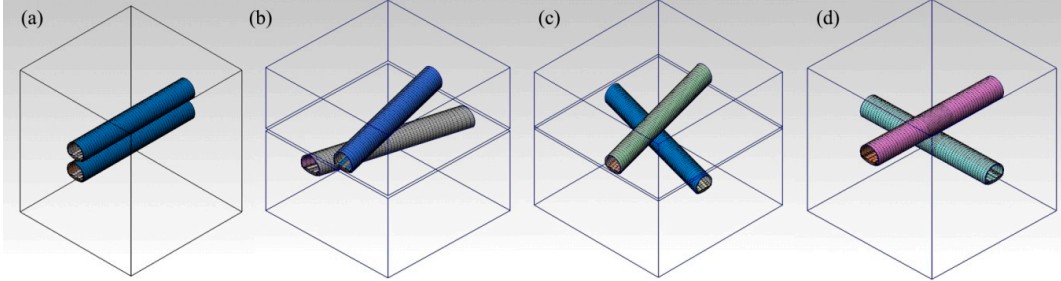

**Figure 2.** Numerical simulation model: the cross angle of two tunnels for (**a**), (**b**), (**c**), and (**d**) are 0°, 30°, 60°, and 90°, respectively.

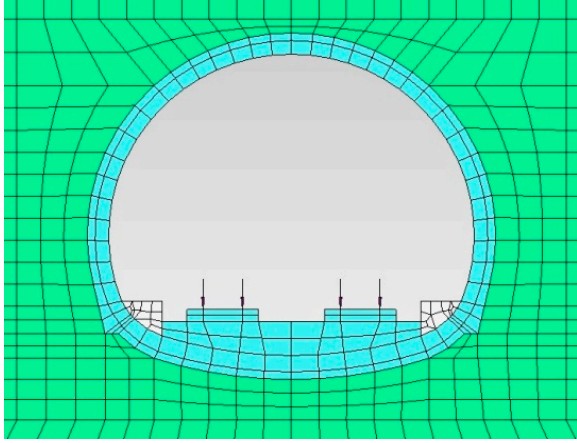

**Figure 3.** Mesh grid.

To determine the static stress and dynamic responses, two phases of the numerical simulation must be undergone. The first phase refers to the simulation of construction excavation under the stratum load, wherein the initial static stresses, which involve the surrounding rock, primary lining, and secondary lining, are determined. A train load is implemented to simulate the tunnel lining response, wherein the dynamic responses, which include the tensile and compressive stresses of the secondary lining, are obtained.

The stress distribution of the tunnel lining was the focus of this paper. Therefore, in the first stage, an equivalent simulation method was adopted to simulate construction excavation, which would simplify the difficulty and computational complexity of the numerical simulation. The detail components of a steel arch model, including the primary lining [24] and rock bolt in surrounding mass [25–27], mainly consist of surrounding rock, primary lining, and secondary lining. The surrounding rock was divided into eight-node solid units, and the material properties were described by the elastoplastic model and the Mohr–Coulomb yield criterion. The reinforcement effect of an anchor rod on surrounding rock was simulated by improving the mechanical parameters of the surrounding rock in the anchor zone (Table 1). The initial support, secondary lining, track plate, concrete support layer, and inverted arch filling layer were divided into eight-node solid units and simulated by linear elastic materials (Table 2).

**Table 1.** Stratum mechanical parameters.

| Rock Level | $\gamma$/kN·m$^{-3}$ | $E$/GPa | $\mu$ | $C$/kPa | $\Phi$/° |
|:---:|:---:|:---:|:---:|:---:|:---:|
| III | 25 | 13.5 | 0.30 | 763.4 | 39 |
| IV | 22 | 3.1 | 0.35 | 153.4 | 30 |
| V | 19.5 | 0.9 | 0.35 | 93.8 | 25 |

**Table 2.** Structural mechanical parameters (e.g., lining and track plates).

| Structural Component | $\gamma$/kN·m$^{-3}$ | $E$/GPa | $\mu$ | $f_t$/MPa | $f_c$/MPa |
|:---:|:---:|:---:|:---:|:---:|:---:|
| Secondary lining (C35) | 26.3 | 31.5 | 0.2 | 1.65 | 17.5 |
| Concrete foundation (C30) | 25 | 30 | 0.2 | 1.5 | 11.0 |
| Filling layer (C25) | 23 | 28 | 0.2 | 1.3 | 12.5 |
| Primary lining (C25) | 25 | 33.4 | 0.2 | 1.3 | 12.5 |
| Track plate (C45) | 27 | 33.5 | 0.2 | 1.9 | 21.5 |

The surrounding rock was simulated by a spring, and the boundary conditions were set (Figure 4). The outer nodes of the spring set the degrees of freedom (DOF) constraints in each direction. The nodes and lining in the spring were set as binding constraints. The DOF constraints were set at both ends

of the model. The viscous damper was arranged in the normal and tangential boundaries, and the normal damping ratio was 1.0. The tangential damping ratio for absorbing the dynamic wave was 0.5.

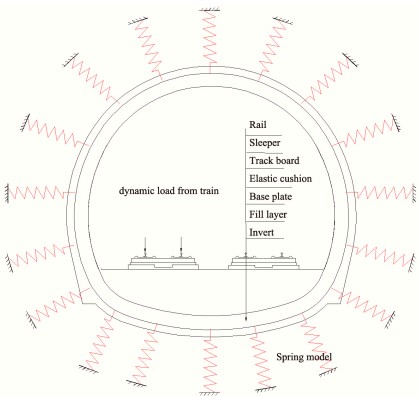

**Figure 4.** Boundary condition.

Before the dynamic calculation, the initial stress was already available in the tunnel lining and the surrounding rock. Thus, the initial stress field and the construction excavation process were simulated first to obtain the initial stress field before the dynamic analysis. Next, the modal analysis was carried out, and the natural vibration frequency of the tunnel's surrounding rock system was calculated to determine the damping coefficient. The vibration load of the train was applied to calculate the structure and the dynamic response of the surrounding rock. The specific calculation steps are presented as follows:

(1) Establish the geometric model of the tunnel and the surrounding rock, set the unit type, and divide the grid.

(2) Set the static boundary and the balance of the initial stratum stress balance to obtain the original rock stress before excavation.

(3) Simulate tunnel excavation. This process was simulated by releasing the stratum load (JTG/T D70-2010). The stratum load was released in two stages. The first stage corresponded to the construction of tunnel excavation and the initial support, and the second stage corresponded to the second lining construction. The released ratio varied with the rock mass grade (Table 3).

**Table 3.** Load-sharing ratio during tunnel construction.

| Rock Level | Primary Lining | Secondary Lining |
|:---:|:---:|:---:|
| III | 100 | 0 |
| IV | 60 | 40 |
| V | 30 | 70 |

(4) Set the material damping and dynamic boundaries, and execute the calculation to obtain the dynamic response from the train load.

### 2.1.2. Train Load

The train–tunnel system is a complex system that includes coupling, time varying, and nonlinear characteristics. Two independent subsystems, namely, the train and the tunnel subsystem, were established. The train–tunnel system was coupled by the geometrical compatibility of the track displacement at the wheel–rail contact with the vehicle displacement, as well as the equilibrium condition of the interaction force between the wheel and the rail. The high-speed train load acted directly on each node of the tunnel structure subsystem.

Two main methods, namely, excitation force function and field measurement, were used to determine the train load in numerical simulation. The excitation force function, which has been proven

feasible by many scholars [28], was selected to be implemented on the tunnel in this paper. In studying the dynamic response characteristics of tunnel structures under train load, the surrounding rock was regarded as a uniform elastic half space, and spring unit simulation was used. The train load was fed into the tunnel subsystem as an external force.

The expression of the train load was used as the excitation model [29]:

$$P(t) = k_1 k_2 (P_0 + P_1 \sin \omega_1 t + P_2 \sin \omega_2 t + P_3 \sin \omega_3 t) \tag{1}$$

where $k_1$ and $k_2$ are the adjacent wheel–rail force superposition coefficient and the dispersion coefficient, which were 1.2–1.7 and 0.6–0.9, respectively; $P_0$ is the vehicle static load; $P_1$, $P_2$, and $P_3$ are the vibration loads. The train mass is set as $M_0$. The vibration load amplitude is

$$P_i = M_0 a_i \omega_i{}^2 \tag{2}$$

where $\omega_i$ is the track irregularity rise; $\omega_i = 2\pi v / L_i$; $v$ is the high-speed train speed; and $L_i$ is the typical wavelength of the geometric irregularity curve.

In this paper, the train axle weight was 17 t, and the mass under the lower spring mass was 750 kg. The control value of wavelength ($L_i$) and vector height ($a_i$) for track irregularity control value were defined in accordance with code TB10003-2005—that is, $L_1$ = 10 m, $L_2$ = 2 m, and $L_3$ = 0.5 m; and $a_1$ = 3.5 mm, $a_2$ = 0.4 mm, and $a_3$ = 0.08 mm, respectively. The history curve of the train load is shown in Figure 5, where train speed $v$= 350 km/h.

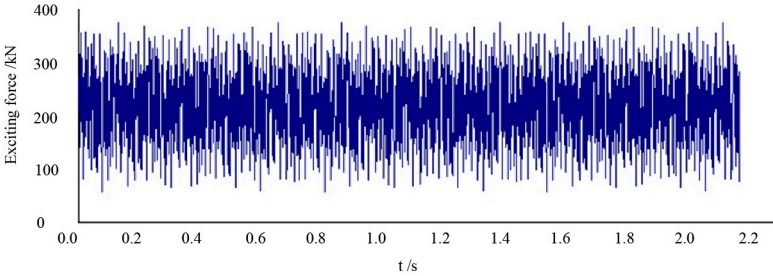

**Figure 5.** Time-history curve of the train load.

*2.2. Fatigue Experiment*

The fatigue experiment aimed to distinguish whether the high lining stress of the cross tunnel will decrease the service life, and how to determine the design parameters of the cross tunnel to maintain the lining stress within a reasonable range. Therefore, a self-made device was utilised to test the characteristics of fatigue evolution under different stress levels.

2.2.1. Specimen Description

In this paper, all specimens were cast using ordinary Portland cement No. 42.5 (R) in the laboratory. Fine aggregate refers to two kinds of granular quartz sand, namely 10–20 mesh and 40–70 mesh; apparent density is 2650 kg/m³; coarse aggregate corresponds to a limestone crushed stone with 5–20 mm particle size. The additive was a polycarboxylic acid water-reducing agent. The mix proportions by weight were presented as follows: 1 cement, 2.697 sand, 3.881 gravel, 0.390 fly ash, and 0.013 additive. The water/cement ratio was 0.552. The average compressive strength (*C*) of the concrete, using a standard 100 mm cube according to GB50080-2002, at 28 d was 42.3 MPa. The size of the fatigue specimens was cuboid (100 mm × 100 mm × 300 mm). Before the test, the dynamic elastic modulus (*E*) and the shear modulus (*G*) were calibrated by using an ETM (Emodumeter-TM) for a non-destructive test. The average dynamic elastic modulus (*E*) and the shear modulus (*G*) were 41.8 and 16.9 GPa, respectively (Table 4).

**Table 4.** Dynamic and static mechanical parameters.

| Dynamic | | | | Static | | | |
| --- | --- | --- | --- | --- | --- | --- | --- |
| Size | E (GPa) | G (GPa) | $v$ | Size | E (GPa) | C (MPa) | $v$ |
| 100 × 100 × 300 mm | 41.8 | 16.9 | 0.23 | 100 × 100 × 100 mm | 31.7 | 42.3 | 0.23 |

### 2.2.2. Fatigue Test System

A fatigue device was designed to simulate the fatigue response of the secondary lining under the special stress level, and analyse the fatigue characteristics of tunnel lining under different stress levels and dynamic loads. The schematic of the experiment is shown in Figure 6, and the real device is shown in Figures 7 and 8. The stress state of the tunnel lining structure was applied by spring, and the load action of the train was simulated by the MTS (an American brand of electro-hydraulic, servo-material testing machine) dynamic system of the fatigue tester, which was divided into two parts: static load and dynamic load. The static load was mainly used as the bottom inverse force of the structure—that is, the initial load simulation was applied in a dynamic system; the static load was mainly the train load. The cyclic load simulation was mainly performed on the basis of the initial load. The stress state of the lining structure and the train load were determined by the numerical calculation results.

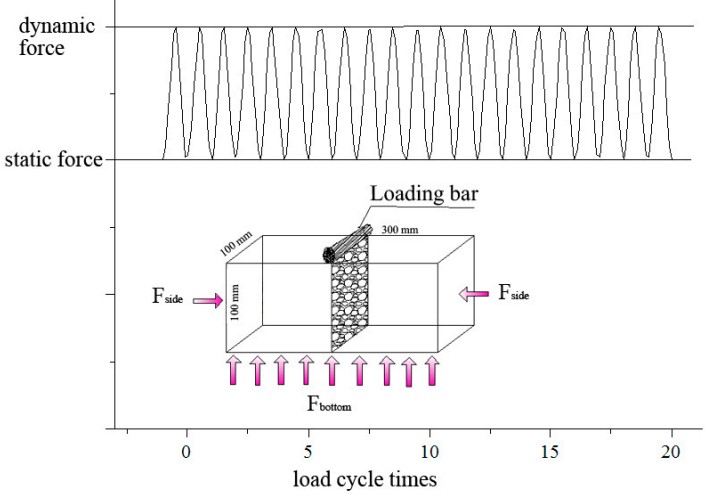

**Figure 6.** Schematic of the fatigue test.

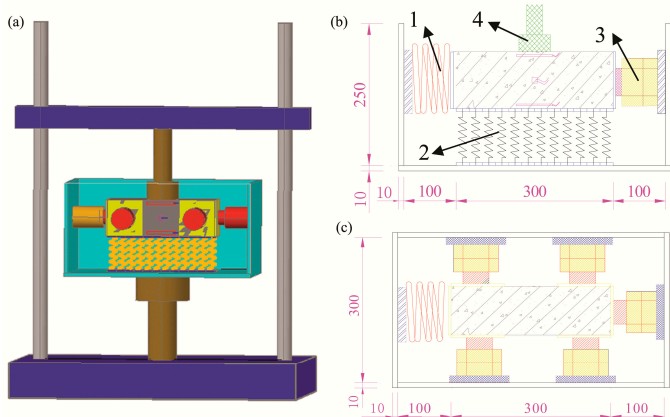

**Figure 7.** Structure of the fatigue test system: (**a**) three-dimensional (3D) model, (**b**) lateral view, and (**c**) top view; 1 and 2 refer to the lateral spring, 3 refers to the hydraulic jacks, and 4 refers to the MTS (American brand of electro-hydraulic, servo-material testing machine).

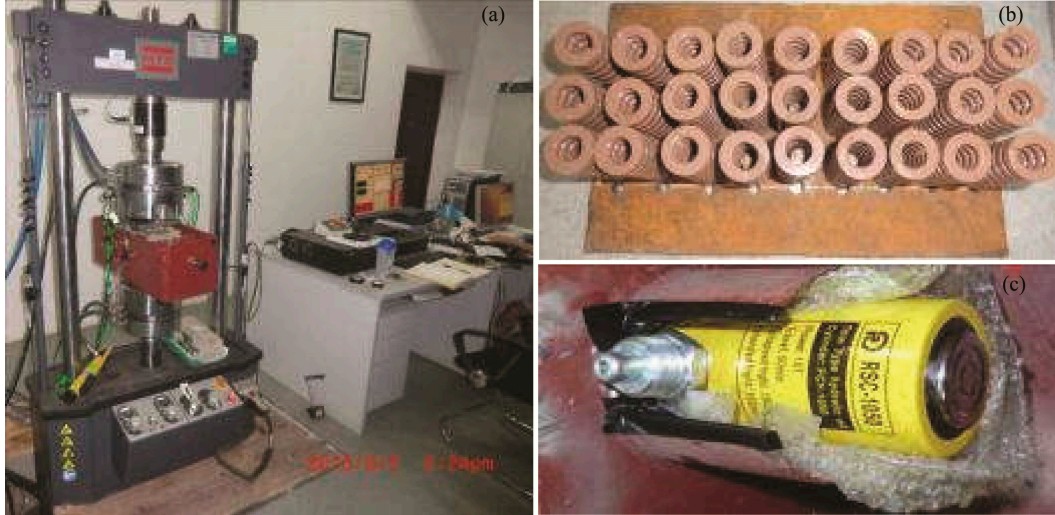

**Figure 8.** Test instrument and its components: (**a**) device box and MTS, (**b**) bottom spring plate, and (**c**) hydraulic jacks.

Fatigue tests were conducted under tensile cyclic loading with their minimum stress levels, according to the initial stress value of cross tunnel and the dynamic amplitude equal to the stress increment, due to the running train. The cycle load was simulated by utilising MTS with a loading frequency of $f_P$ = 10 Hz. The fatigue test was stopped either upon the fracture failure of the specimen or after cyclic number $N = 2 \times 10^6$ load cycles.

The strains were analysed in the area of the concrete fatigue. Therefore, the strains selected in this paper were installed on the specimen surface at the theoretical maximum tensile and compressive zone. The dynamic strain gauge adopted a foil resistance strain gauge, and the test accuracy was 2.18% ± 1%.

### 2.2.3. Test Cases

According to the numerical results of the lining stress level, three influencing factors—namely, cyclic dynamic load, vertical static load, and loading frequency of running train—were considered. The numbers of cycles to failure can show considerable scatter. The evolution characteristics of specimen would also be affected by scatter. Therefore, each influencing factor considered three groups of experimental conditions to preclude the influence of batch on the results. The two factors, namely, static and dynamic load levels, were analysed in the test.

(1) Monotonic test for ultimate capacity: The stress level was selected as the experimental control variable. Five specimens were tested under monotonic loading, until the measurement of the ultimate tensile capacity failed. Under the confining pressure of 1.5 MPa (15 kN lateral force), the penetrating fracture of the specimen is revealed when the static tensile load reaches 41.36 kN (average value of five specimens). The tensile stress generated in the zone was close to the axial tensile strength of the structure $f_{tk}$ = 2.56 MPa.

(2) Static load level test: The static load mainly affects the initial damage of the tunnel structure. Therefore, the static load test of the structure was first performed to obtain the cumulative damage behaviour of the structure under the action of different load levels, as well as the ultimate loads from intact to destructive. The static load threshold value, which causes the fatigue damage of the tunnel lining, was obtained through the fatigue test of the structure under different static load conditions. The test conditions and cases are shown in detail in Table 5. The tensile loading was applied at a loading rate of 1 kN/min. The tensile tests were carried out in accordance with GB50080-2002, which is equivalent to BS EN 13480-1. The load was kept for 3 min, and the specimen was checked for cracks.

**Table 5.** Case for different levels of static load (stress level).

| Parameters | | | Cases | S-1 | S-2 | S-3 | S-4 |
|---|---|---|---|---|---|---|---|
| Lateral load(kN) | Dynamic load amplitude (kN) | Loading frequency (Hz) | Static load force | 25 kN | 27 kN | 29 kN | 31 kN |
| 1.5 | 2.4 | 12 | Stress level | 0.65 | 0.7 | 0.75 | 0.80 |

(3) Dynamic load level test: Expressing the dynamic stress of the tunnel lining is difficult when using simple calculation formulas, because of train axle weight, running speed, track irregularities, and tunnel bottom conditions. Therefore, the values of the dynamic load in this paper are determined on the basis of two factors: (1) full manifestation of the entire evolution process of the specimen fatigue testing, from the initial damage to destruction, and (2) the analysis of field test data and theoretical calculations of the circular railway experimental test [30]. The dynamic stress level of fatigue in this test is shown in Table 6.

**Table 6.** Case for different levels of dynamic load (dynamic stress amplitude).

| Parameters | | | Cases | D-1 | D-2 | D-3 | D-4 |
|---|---|---|---|---|---|---|---|
| Lateral load(kN) | Vertical static load (kN) | Loading frequency (Hz) | Dynamic load amplitude | 1.6 kN | 2.4 kN | 3.6 kN | 4.5 kN |
| 1.5 | 27 | 12 | Stress level | 0.6 | 0.7 | 0.75 | 0.85 |

## 3. Results and Analysis

### 3.1. Stress Level of Secondary Lining

Figures 9 and 10 show the distribution of secondary linings $\sigma_1$ and $\sigma_3$ along the longitudinal direction of the tunnel under the basic working condition, where the red line refers to the static stress without a train, and the blue line corresponds to the dynamic stress, which includes the train load.

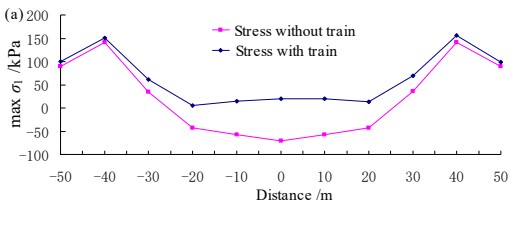
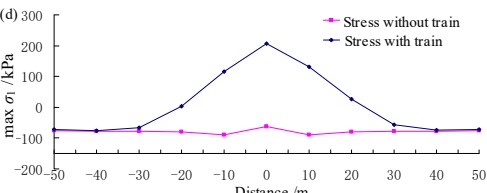
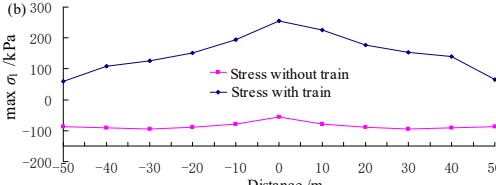
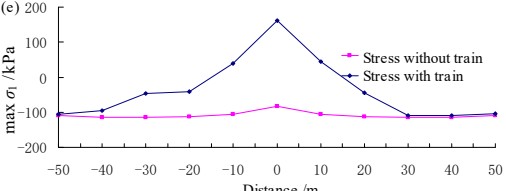

**Figure 9.** *Cont.*

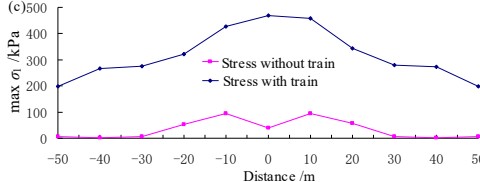 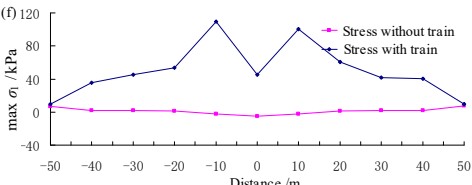

**Figure 9.** Distribution of secondary lining $\sigma_1$ along the longitudinal direction of the tunnel: (**a**) arch of upper tunnel, (**b**) side wall of upper tunnel, (**c**) inversion of upper tunnel, (**d**) arch of lower tunnel, (**e**) side wall of lower tunnel, and (**f**) inversion of lower tunnel.

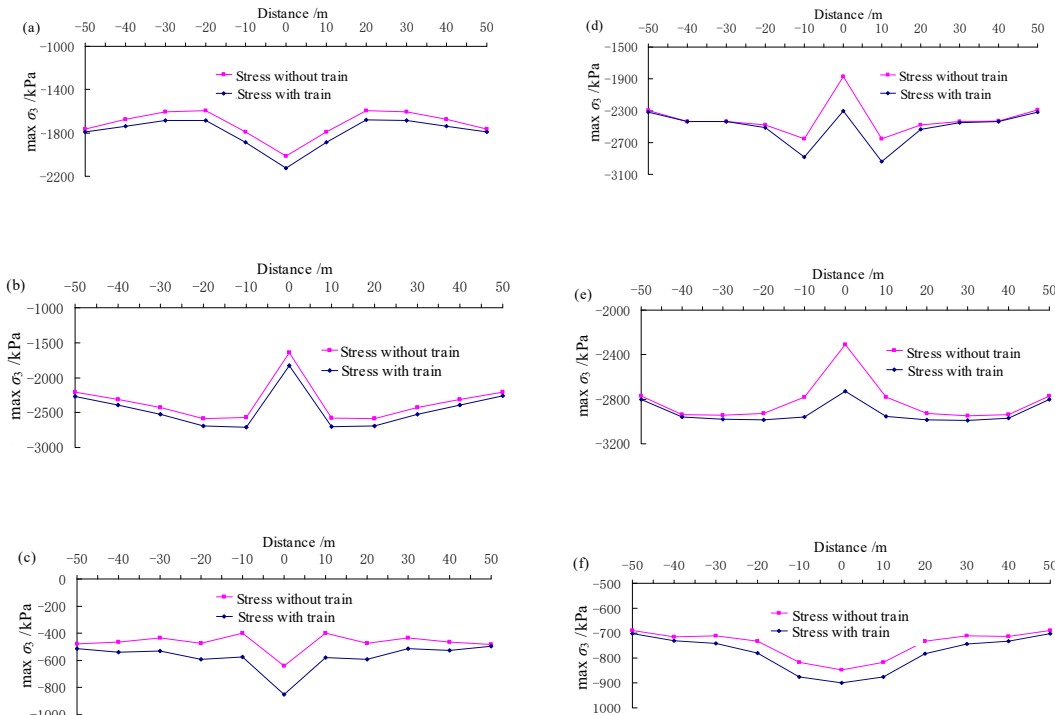

**Figure 10.** Distribution of secondary lining $\sigma_3$ along the longitudinal direction of the tunnel: (**a**) arch of the upper tunnel, (**b**) side wall of the upper tunnel, (**c**) invert of the upper tunnel, (**d**) arch of the lower tunnel, (**e**) side wall of the lower tunnel, and (**f**) invert of the lower tunnel.

Figure 9 shows that the stress value $\sigma_1$ of the invert in the upper tunnel is higher than that of the side wall, and even than that of the arch when no train is passing. The stress value would increase evidently from 142.0 kPa (arch), −54.8 kPa (side wall), and 95.6 kPa (invert) to 156.5 kPa (arch), 255.9 kPa (side wall), and 469.9 kPa (invert) when trains pass. The longitudinal range of increasing stress caused by the tunnel crossing is approximately 20 m (1.5$D$, with $D$ as the diameter of the tunnel) from the intersection. The stress value $\sigma_1$ of the arch in the lower tunnel is higher than that of the side wall, and even than that of the invert. The stress value of the arch increases more significantly for the lower tunnel, and that of the invert does so for the upper tunnel, due to the train loading. The distribution of $\sigma_3$ has similar characteristics (Figure 10)—that is, the invert for the upper tunnel and arch for the lower tunnel have higher stress levels, either due to the just stratum load or the stratum load and the train load (Figure 11). Regardless of initial stress or dynamic stress, the lining concrete in the invert of upper tunnel shows a typical tensile status, and the lining concrete in the arch of the lower tunnel exhibits an evident compressive condition. Therefore, the stress value for the upper tunnel invert is selected as the reference for the fatigue experiment—with initial static stress and dynamic stress the as minimum and maximum stress, respectively—because the lining concrete, which is a brittle material, presents a higher compressive strength than a tensile one.

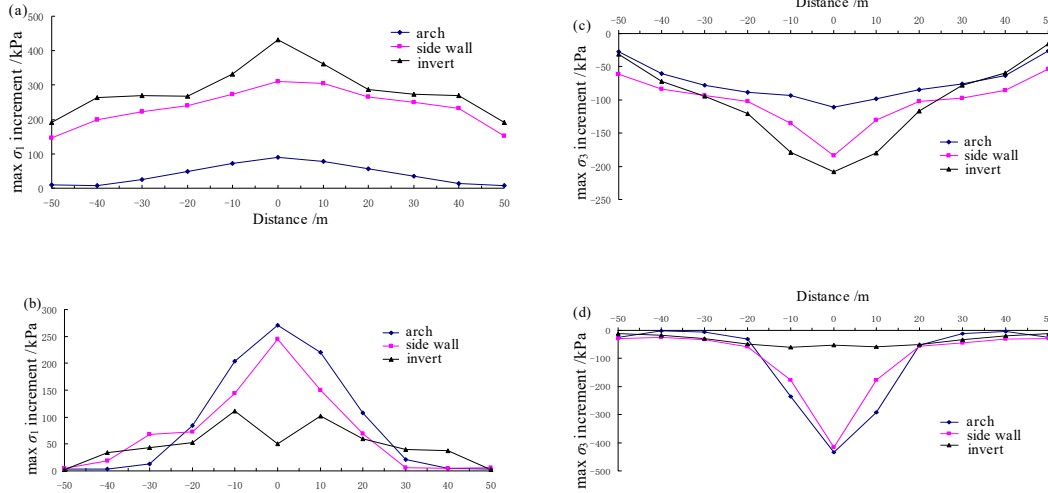

**Figure 11.** Stress increment comparison due to the train loading: (**a**) and (**b**) show the major principal stress of the upper and lower tunnels, respectively; (**c**) and (**d**) illustrate the third principal stress of the upper and lower tunnels, respectively.

Similarly, the static initial tensile stress and the maximum dynamic tensile stress of the tunnel lining under different work conditions were used as input values for the minimum and maximum stress levels of the fatigue loads. Table 7 presents the maximum dynamic tensile stress increment $\Delta\sigma$ under different work conditions. It also shows that the stress increment of the tunnel lining is mostly affected by the way the train passes. Poor surrounding rock level, high train running speed, and thin surrounding rock thickness can also considerably increase the stress increment of the tunnel lining. Moreover, the tunnel lining would suffer a higher increment of dynamic stress, even under the same train load, when the static stress is relatively large due to the stratum. Assuming that the influencing factors are independent of each other, the fitting formula of the maximum increment of the dynamic tensile stress in the lining structure, with regard to the surrounding rock level ($\lambda$), train speed ($v$), rock pillar height ($H$), and cross-angle ($\theta$), can be obtained by considering the condition that the train passes through both the upper and lower tunnels simultaneously. As shown in Equation (3), where $i = 1, 2$, and 3 represents grade III, IV, and V surrounding rock, respectively, $\lambda_1$, $\lambda_2$, and $\lambda_3$ are 0.7, 0.79, and 1.0, respectively.

$$\Delta\sigma_{(\lambda,v,\theta,H)} = 7.9 \times 10^{-6}\lambda_i\left(1.15e^{\frac{v}{150.7}} - 1.75\right)\left(10.53e^{-\frac{H}{23.31}} - 0.152\right)\left(2.62e^{\frac{\theta}{21.33}} + 820.8\right) \tag{3}$$

**Table 7.** Maximum dynamic tensile stress increment for different design parameters of the cross tunnel.

| Condition | | Dynamic Tensile Stress Increment (MPa) | Location |
|---|---|---|---|
| Rock level | III | 0.357 | Side wall of upper tunnel |
| | IV | 0.404 | Invert of upper tunnel |
| | V | 0.513 | Invert of upper tunnel |
| The way the train passes | Upper tunnel | 0.513 | Invert of lower tunnel |
| | Lower tunnel | 0.386 | Side wall of upper tunnel |
| | Both tunnels simultaneously | 0.789 | Invert of lower tunnel |
| Train speed | 250 km/h | 0.221 | Invert of upper tunnel |
| | 300 km/h | 0.343 | Invert of upper tunnel |
| | 350 km/h | 0.513 | Invert of upper tunnel |

**Table 7.** *Cont.*

| Condition | | Dynamic Tensile Stress Increment (MPa) | Location |
|---|---|---|---|
| Cross-angle | 0° | 0.378 | Invert of upper tunnel |
| | 30° | 0.385 | Invert of upper tunnel |
| | 60° | 0.398 | Invert of upper tunnel |
| | 90° | 0.513 | Invert of upper tunnel |
| Rock pillar thickness between two tunnels | 1 m | 0.513 | Invert of upper tunnel |
| | 3 m | 0.461 | Invert of upper tunnel |
| | 5 m | 0.432 | Invert of upper tunnel |
| | 10 m | 0.348 | Invert of upper tunnel |

### 3.2. Evolutionary Characteristics of Dynamic Strain

A failing structure caused by progressive fatigue degradation usually depends on the fatigue loading rate, the fatigue stress ratio $f_{min}/f_{max}$ (where $f_{min}$ represents the minimum tensile stress under static load and $f_{max}$ represents the maximum tensile stress under dynamic load), and the number of cycles. Given a specific loading rate, the stress ratio $f_{min}/f_{max}$ directly determines the service life of the structure.

#### 3.2.1. Fatigue Evolution Model

The S-shaped curve of fatigue strain can be divided into three stages: the first stage is a disproportionate increase in deformation; the second stage is a linear increase in deformations; and the third stage is a destruction stage, caused by the unstable increase in the crack. Considering the initial strain, instability rate, and fatigue life, the structural strain evolution model can reflect the three-stage strain characteristics of the structure under cyclic loading, and describe the evolution of material fatigue strain quantitatively [31]. The specific strain evolution equation is expressed as follows:

$$\frac{\varepsilon_n}{\varepsilon_0} = \frac{\varepsilon_1}{\varepsilon_0} + \alpha \left( \frac{\beta}{\beta - N/N_f} \right)^{\frac{1}{\rho}} \tag{4}$$

where $\varepsilon_0$, $\varepsilon_1$, and $\varepsilon_n$ are the initial, first, and cumulative strains, respectively; $N$ and $N_f$ are the number of cycles and fatigue life, respectively; the ratio $N/N_f$ is the cycle ratio; and $\alpha$, $\beta$, and $\rho$ are the evolution equation parameters: $\alpha$ refers to the instability scale factor, $\beta$ refers to the instability factor, and $\rho$ refers to the instability velocity factor.

The instability velocity factor $\rho$, related to the convergence speed of the strain evolution curve, is generally recommended to be in the range of 2–8. The instability scale factor $\alpha$, which characterises the ratio of the failure stage in the structural fatigue life curve, is generally recommended to be in the range of $0$–$(1 - \varepsilon_1/\varepsilon_0)$. The relationship of the instability factor $\beta$ is related to $\alpha$ and $\rho$ when $N = N_f$ is presented as follows:

$$\beta = \left( \frac{1 - \varepsilon_1/\varepsilon_0}{\alpha} \right)^{-\rho} + 1 \tag{5}$$

#### 3.2.2. Strain Curve at Different Static Forces

Figure 12 shows the dynamic strain evolution curves of the tunnel lining under four static load levels. The following remarks can be observed from Figure 12.

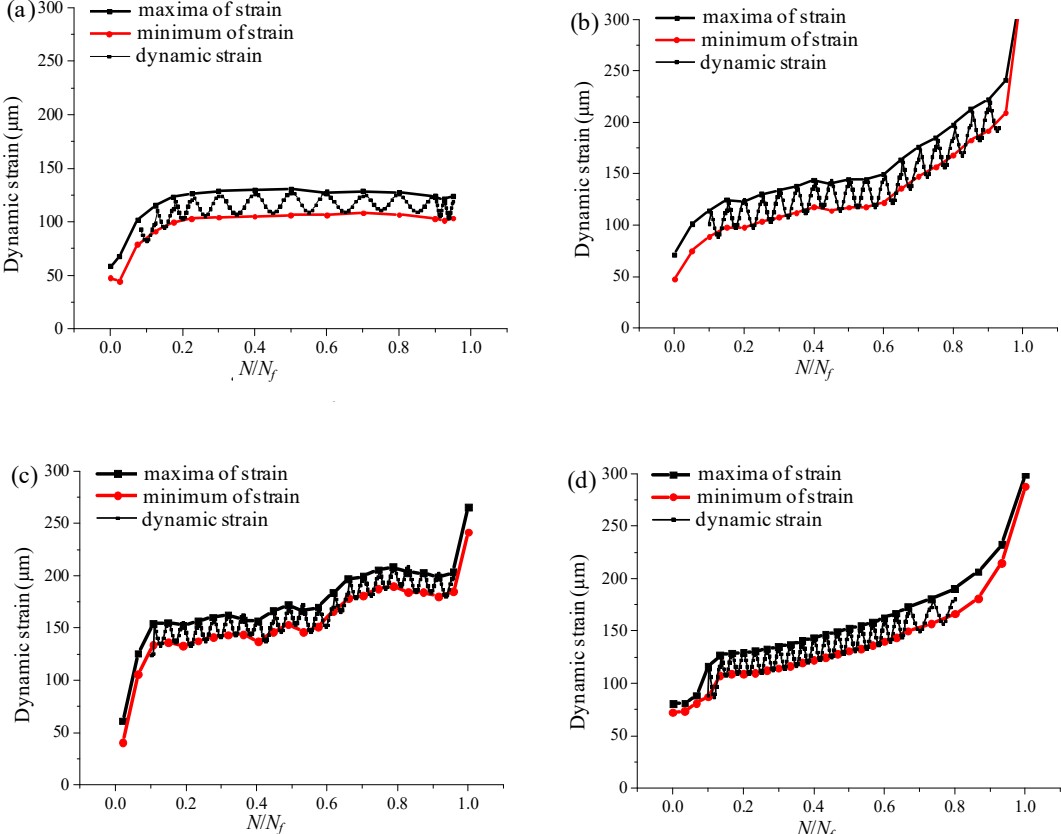

**Figure 12.** Dynamic strain curve: (**a**) static load = 25 kN, (**b**) static load = 27 kN, (**c**) static load = 29 kN, and (**d**) static load = 31 kN.

The strain cumulative evolution curve presents the typical three-stage characteristics: a disproportionate increase in deformation, a linear increase in deformations, and a destruction stage due to the unstable increase in crack; the test results agree with the theoretical curve overall. The evident increase occurs in the first and third stage, due to the deformation of microscopic cracks for the former and the rapid increase of macroscopic fracture until failure for the latter. The dynamic strain of the tunnel lining increases with the cyclic ratio ($N/N_f$) and shows the characteristics of nonlinear enlargement in the second stage.

The fatigue failure of the lining structure is closely related to the stress level. When the static load stress level is less than 0.6, no failure stage or material damage occur in the structural strain process (Figure 12a). When the static load stress level is greater than 0.6, the entire strain growth process of the structure can be divided into three different stages. The change in structural strain is relatively stable in the middle stage, the growth of structural strain is relatively fast in the initial stage, while the structural strain increases evidently and structural damage occurs during the failure stage (Figure 12b–d).

In the first stage of specimen failure, no visible cracks or evident deformations can be observed on the surface of the structure, indicating that the evolution of structural damage is the development process of the defects, such as micropores and microcracks inside the material, which conform to the irreversible cumulative damage and failure characteristics of the structure. The appearance of these phenomena indicates that the specimen firstly develops the initial crack in the lower tensile region. After a long period of damage accumulation, the microcrack runs through and rapidly expands into a macroscopic crack, which eventually forms a bottom–up failure crack (Figure 13).

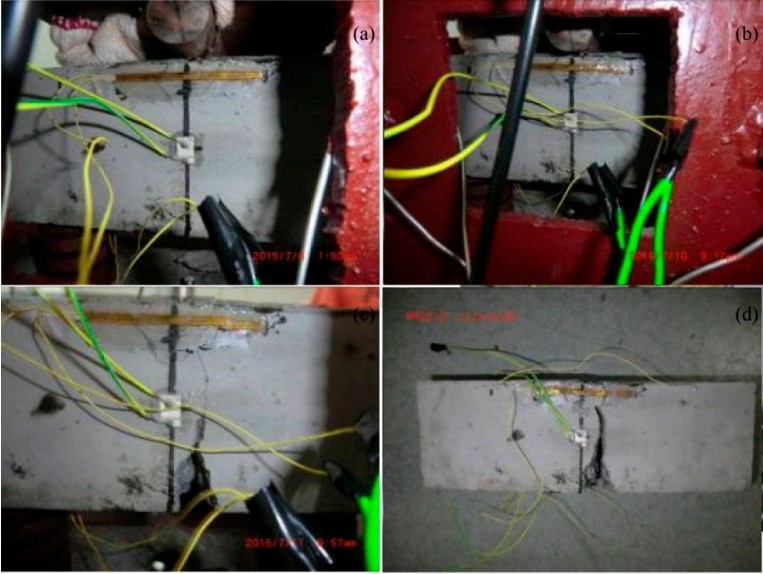

**Figure 13.** Damage state of the inverted arch structure at different stages: (**a**) initial stage, (**b**) intermediate stage, (**c**) failure stage, and (**d**) cracking position.

### 3.2.3. Strain Curve at Different Loading Ratios

Figure 14 shows a test curve for the dynamic strain evolution characteristics of structural specimens under four dynamic load conditions. The results show that under low dynamic stress amplitude, the variation value of strain amplitude and strain ratio becomes stable after the initial growth, and the structure may not exhibit fatigue failure. Figure 14a shows that at the dynamic stress level of 1.6 kN, the number of load cycles reaches $2 \times 10^6$ times, the specimen remains undamaged, and the change value of the strain of the specimen shows a stable state.

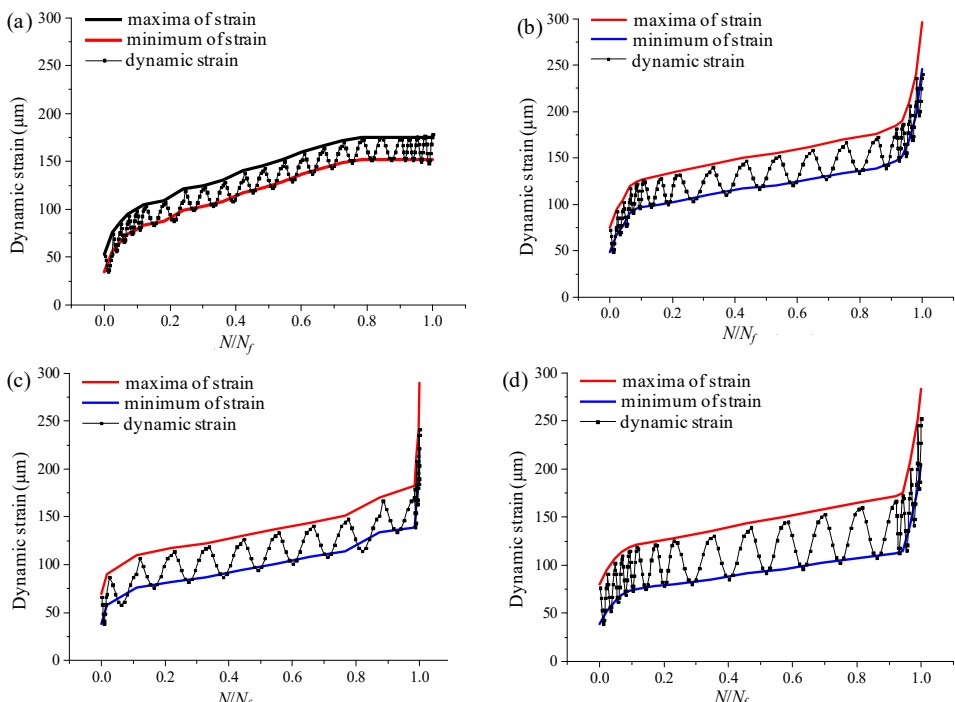

**Figure 14.** Dynamic strain curve: (**a**) dynamic load = 1.6 kN, (**b**) dynamic load = 2.4 kN, (**c**) dynamic load = 3.6 kN, and (**d**) dynamic load = 4.5 kN.

The fatigue failure characteristics for high dynamic stress amplitude are substantial. The number of load cycles that correspond to structural failure decreases that occur gradually with the increase in the dynamic stress amplitude. For example, under the conditions of dynamic stress amplitudes of 2.4, 3.6, and 4.5 kN, the number of cyclic load vibrations that correspond to structural failure is approximately $7.96 \times 10^5$, $2.15 \times 10^5$, and $1.54 \times 10^4$, respectively (Figure 14b–d).

### 3.3. S–N Curve

Based on the structural strain evolution model, combined with the measured data (Figures 15 and 16), the fitting parameters of different working conditions are obtained as shown in Table 8.

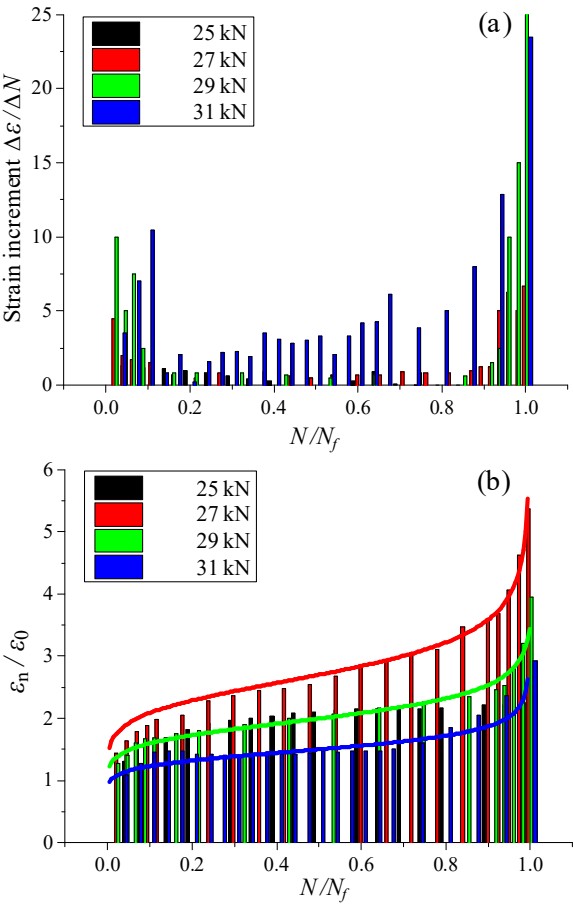

**Figure 15.** Evolution of strain characteristics for different static stress levels: (**a**) strain increment and (**b**) $\varepsilon_n/\varepsilon_0$.

**Table 8.** Strain evolution equation fitting parameters.

| Fitting Equation | Fitting Parameters | $\varepsilon_1/\varepsilon_0$ | $\alpha$ | $\beta$ | $\rho$ |
|---|---|---|---|---|---|
| $\frac{\varepsilon_n}{\varepsilon_0} = \frac{\varepsilon_1}{\varepsilon_0} + \alpha\left(\frac{\beta}{\beta-N/N_f}\right)^{\frac{1}{\rho}}$ | Dynamic load = 2.4 kN | 0.7 | 1.6 | 1.01 | 5.5 |
| | Dynamic load = 3.6 kN | 0.7 | 1.8 | 1.01 | 5.0 |
| | Dynamic load = 4.5 kN | 0.7 | 1.0 | 1.00 | 4.0 |
| | Static load = 27 kN | 0.7 | 2.0 | 1.00 | 6.0 |
| | Static load = 29 kN | 0.6 | 1.4 | 1.01 | 6.5 |
| | Static load = 31 kN | 0.5 | 1.0 | 1.00 | 5.0 |

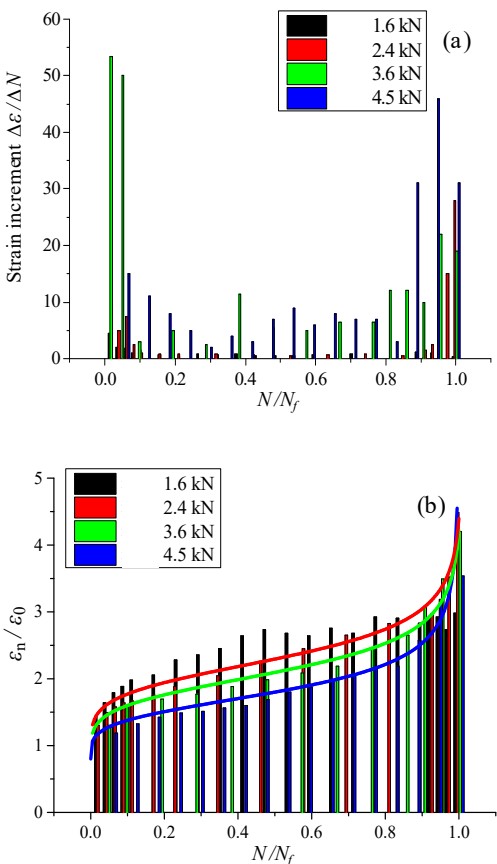

**Figure 16.** Evolution of strain characteristics for different dynamic stress: (**a**) strain increment, (**b**) $\varepsilon_n/\varepsilon_0$.

The relationship curve between structural strain $\varepsilon$ and number of cycles $N$ was extracted to describe the entire process of structural strain development under cyclic loading (Figure 17). The test results show that the strain development process only included the first two stages of fatigue failure, and no failure stage when the stress level was less than 0.6. The strain development process involves all three phases, namely, a disproportionate increase in deformation, a linear increase in deformations, and a destruction stage when the stress level is greater than 0.6. Moreover, the ultimate load cycles were less than $2 \times 10^6$. Table 9 presents the results of $N_f$ for different stress levels $f_{max}/f_t$.

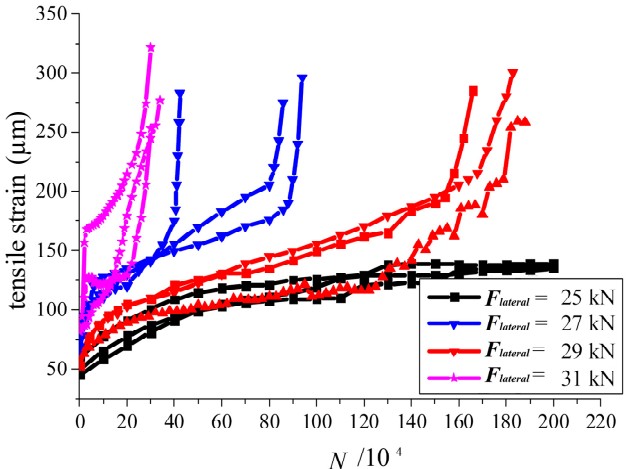

**Figure 17.** Process of strain evolution.

**Table 9.** Relationship between the maximum tensile stress under dynamic load ($f_{max}$)/$f_t$ and fatigue life ($N_f$) from the fatigue test.

| $f_{max}/f_t$ | $f_{min}/f_{max}$ | $N_f$ ($10^6$) |
| --- | --- | --- |
| 0.6 | 0.68 | >2 |
| 0.65 | 0.63 | 2.03, 1.62, 1.18 |
| 0.7 | 0.58 | 0.94, 0.863, 0.72 |
| 0.75 | 0.56 | 0.423, 0.301, 0.38 |

The relevant parameters of fatigue life can be obtained in Table 9 based on the data of the fatigue test under different strain levels; the *S*–*N* curve (Figure 18) describes the stress level, and fatigue life is obtained and expressed by the following linear fitting equation:

$$S = -0.145lgN + 0.975, \ (r = 0.912) \tag{6}$$

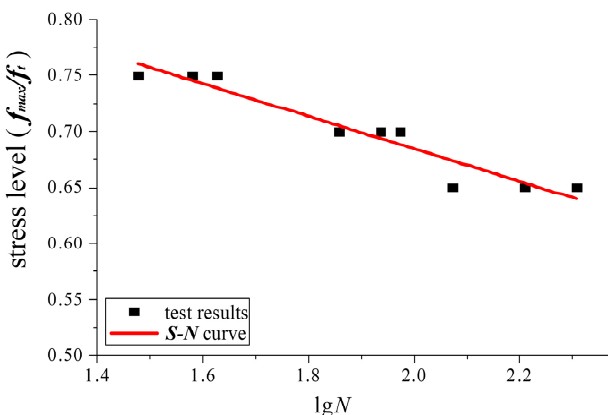

**Figure 18.** Least squares fitting of *S*–*N* curve fitting.

## 4. Stress Influence Zoning

### 4.1. Method of Influence Zoning

According to the above research, the influence zoning method of cross tunnel is proposed to evaluate its practicing design parameters, based on the fatigue effect due to train loading:

#### 4.1.1. Zoning Criteria

The design life of an HSR, which belongs to the problem of stress-controlled fatigue, is 100 years. The zoning criteria are proposed on the basis of two indicators: namely, the initial static tensile stress $f_{min}$ and the maximum dynamic tensile stress $f_{max}$, as follows:

Criterion 1 (initial stress): The initial static tensile stress of the secondary lining concrete $f_{min}$ should not exceed 0.6 times the design strength under stratum loading [32].

Criterion 2 (dynamic stress): The dynamic tensile stress $f_{max}$ should not exceed the value that corresponds to $N_f = 2.0 \times 10^6$, which can be calculated by Equation (6).

#### 4.1.2. Zoning Method

The initial stress state and the maximum dynamic tensile stress are preliminarily determined according to the design parameters of a cross tunnel, and were compared with the discriminating Criteria 1 and 2, respectively, as follows:

(1) If a cross tunnel exceeds both Criteria 1 and 2, then the fatigue damage of secondary lining for these cross tunnels would be very substantial under train loads, such that their service life should be less than the design life. This condition is defined as strong influence.

(2) If a cross tunnel exceeds Criterion 1 but does not Criterion 2, then some minor fatigue damage of secondary lining may occur for these cross tunnels under train load. This condition is defined as weak influence.

(3) If a cross tunnel meets Criteria 1 and 2, then that tunnel structure can be considered to avoid suffering from fatigue damage under the action of train load. This condition is defined as no influence.

### 4.1.3. Zoning Procedure

The specific process of influence zoning is shown in Figure 19.

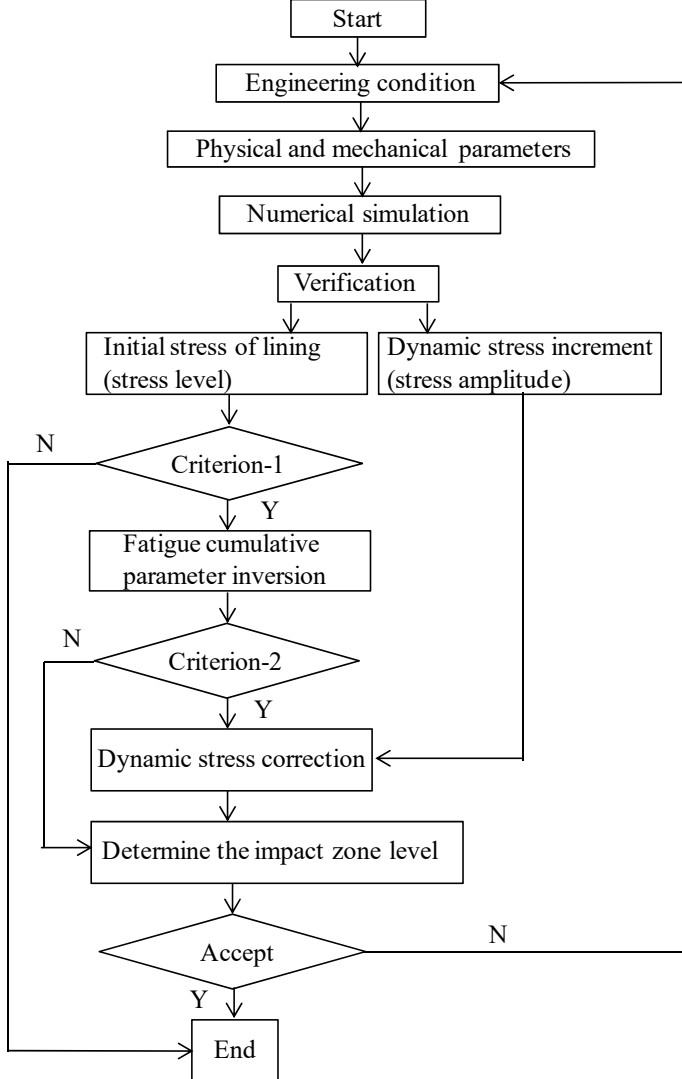

**Figure 19.** Influential zoning procedure of cross tunnel.

### 4.2. Application of Influence Zoning of HSR Cross Tunnels

According to the above method and Equation (3), considering the four parameter variables of surrounding rock grade, cross angle, train running speed, and rock pillar height, the cross tunnel influence zoning criteria are obtained as shown in Tables 10–12. In tables, *D* represents the cross-sectional span of the tunnel. It is shown from Table 11 that when the height of the rock pillar (*H*) between the upper and lower tunnels is less than 3 m and the level IV surrounding rock, the tunnel lining may have a significantly strong influence area.

**Table 10.** Influence zoning of high-speed railway (HSR) cross tunnels for train speed = 250 km/h.

| Rock Level | Angle/° | Rock Thickness | | |
|---|---|---|---|---|
| | | Strong | Weak | None |
| III | 0–60 | - | $H \leq 0.2D$ | $H > 0.2D$ |
| | 60–90 | - | $H \leq 0.5D$ | $H > 0.5D$ |
| IV | 0–60 | - | $H \leq 0.4D$ | $H > 0.4D$ |
| | 60–90 | - | $H \leq 0.7D$ | $H > 0.7D$ |
| V | 0–60 | - | $H \leq 0.8D$ | $H > 0.8D$ |
| | 60–90 | - | $H \leq 1.1D$ | $H > 1.1D$ |

**Table 11.** Influence zoning of HSR cross tunnels for train speed = 300 km/h.

| Rock Level | Angle/° | Thickness of Rock Pillar | | |
|---|---|---|---|---|
| | | Strong | Weak | None |
| III | 0–60 | - | $H \leq 0.8D$ | $H > 0.8D$ |
| | 60–90 | - | $H \leq 1.1D$ | $H > 1.1D$ |
| IV | 0–60 | - | $H \leq 1D$ | $H > 1D$ |
| | 60–90 | $H \leq 0.2D$ | $0.2D < H \leq 1.3D$ | $H > 1.3D$ |
| V | 0–60 | $H \leq 0.4D$ | $0.4D < H \leq 1.5D$ | $H > 1.5D$ |
| | 60–90 | $H \leq 0.7D$ | $0.7D < H \leq 1.7D$ | $H > 1.7D$ |

**Table 12.** Influence zoning of HSR cross tunnels for train speed = 350 km/h.

| Rock Level | Angle/° | Rock Thickness | | |
|---|---|---|---|---|
| | | Strong | Weak | None |
| III | 0–60 | $H \leq 0.5D$ | $0.5D < H \leq 1.6D$ | $H > 1.6D$ |
| | 60–90 | $H \leq 0.7D$ | $0.7D < H \leq 1.8D$ | $H > 1.8D$ |
| IV | 0–60 | $H \leq 0.7D$ | $0.7D < H \leq 1.8D$ | $H > 1.8D$ |
| | 60–90 | $H \leq 1D$ | $1D < H \leq 2D$ | $H > 2D$ |
| V | 0–60 | $H \leq 1.1D$ | $1.1D < H \leq 2.2D$ | $H > 2.2D$ |
| | 60–90 | $H \leq 1.4D$ | $1.4D < H \leq 2.4D$ | $H > 2.4D$ |

## 5. Conclusions

Aiming at the typical engineering weak link of the cross tunnels of HSRs, this paper analyses the initial stress state of the intersection and the dynamic response characteristics of a structure under train load through numerical calculation and indoor fatigue test. This paper subsequently presents a method to determine the influence zoning of vibration. The following main conclusions are obtained through the aforementioned analysis:

(1) The initial stress state and dynamic response at the cross point are substantially larger than that of an ordinary high-speed tunnel structure, especially the local position of the lining structure in the 20 m range near the intersection point. The initial stress state and dynamic response reach the level of the heavy-duty railway substrate.

(2) The invert for the upper tunnel and arch for the lower tunnel have a higher stress level, either due to the just stratum load or the stratum load and the train load. Regardless of initial stress or dynamic stress, the lining concrete in the invert of upper tunnel shows a typical tensile status, and the lining concrete in the arch of the lower tunnel exhibits an evident compressive condition.

(3) The increase in the initial stress state and tensile stress shows a coupling effect. A high initial stress state and tensile stress increment may greatly reduce the durability of the lining structure.

(4) When the height of the rock pillar between the upper and the lower tunnels is less than that of the 3 m and the Grade IV surrounding rock, the tunnel lining may have a considerably strong influence

area—that is, the service life of the invert of the upper tunnel under the train load may be lower than the design requirements.

In this paper, a method is proposed to determine zoning based on numerical simulation and indoor model test. The actual stress state of a cross tunnel is affected by many factors, such as construction method and reserved settlement. Its calculated impact zoning should be further compared and analysed according to measured results in the field.

**Author Contributions:** W.Y., E.D., N.L. and R.F. conceived and designed the experiments; C.S., W.Y. and H.Y. analyzed the data; E.D. and W.Y. wrote the paper. All authors have read and agreed to the published version of the manuscript.

**Funding:** This research was funded by [National Natural Science Foundation of China] grant number [51978670] and [the Fundamental Research Funds for the Central Universities of Central South University] grant number [2019zzts291].

**Conflicts of Interest:** The authors declare no conflict of interest.

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
