# Peer review of "Lining Fatigue Test and Influence Zoning of Tridimensional Cross-Tunnel under High-Speed Train Loads"

_applsci, doi:10.3390/app10165694_

Round 1

Reviewer 1 Report

Overall, I liked the article very much. The subject matter is very important in the context of the existing high-speed railway and their development.
I have included some remarks and comments in the manuscript.

Author Response

The following is our response to the reviewer's comments one by one. The corresponding revisions have been highlighted in yellow in the revised manuscript.

1.There is a presentation problem on line 49.We have rewritten the sentence as "Other investigations also presented that the stress level of the tunnel lining and the surrounding rock in an intersection is generally higher than that in other locations".

2.On line 61, we miswrote the word 'mass' as' mess '. The text has been modified one by one according to the comments of reviewers.

3. On line 113, the mohr-coulomb yield criterion is used to describe the state of soil. Relevant parameters are shown in Table 1. When element stress reaches the yield criterion, the element could be marked as damage. While, the yield element will not be deleted when calculating.

4. Figure 4. The boundary conditions mainly affect the deformation of the lining structure under the action of the released surrounding rock load. In the numerical simulation of this paper, the stratum load was released in 2 stages. The first stage corresponded to the construction of tunnel excavation and the initial support, and the second stage corresponded to the second lining construction. The released ratio varied with the rock mass grade.  

5. Line 138, We have made a revision according to the instruction of the reviewer.

6. In this paper, the structural response of the tunnel is mainly considered. The crack zone of surrounding rock is equivalent by appropriate reduction of surrounding rock mechanical parameters. As shown in Table 1. 

7. Figures 15 and 16. We have reedited the figures according to the instruction of the reviewer.

8. Figure 19. We have reedited the flowchart according to the instruction of the reviewer.

Reviewer 2 Report

The research is very well made and the paper has merit.

The paper addresses a key aspect as regards high speed railway tunnel construction: when a high speed railway tunnel passes over another, there is a concentration of dynamic loads which may cause long term fatigue damages. In the paper, a new methodology to modelize the dynamic response characteristics of a structure under train load, and to evaluate such damages is developed. The paper is well written and easy to read, and the conclusions are consistent with the methodology results.

The authors should correct several typos: there are some traces of the word corrector which were not removed, and some words have been left underlined in yellow.

Author Response

Thank you reviewers for your careful comments. We have checked all the words in the manuscript as requested by the reviewer. The yellow highlighted text has also been modified accordingly.